# OPTIMAL ATTACKS AGAINST MULTIPLE CLASSIFIERS

## ABSTRACT

We study the problem of designing provably optimal adversarial noise algorithms that induce misclassification in settings where a learner aggregates decisions from multiple classifiers. Given the demonstrated vulnerability of state-of-the-art models to adversarial examples, recent efforts within the field of robust machine learning have focused on the use of ensemble classifiers as a way of boosting the robustness of individual models. In this paper, we design provably optimal attacks against a set of classifiers. We demonstrate how this problem can be framed as finding strategies at equilibrium in a two player, zero sum game between a learner and an adversary and consequently illustrate the need for randomization in adversarial attacks. The main technical challenge we consider is the design of best response oracles that can be implemented in a Multiplicative Weight Updates framework to find equilibrium strategies in the zero-sum game. We develop a series of scalable noise generation algorithms for deep neural networks, and show that it outperforms state-of-the-art attacks on various image classification tasks. Although there are generally no guarantees for deep learning, we show this is a well-principled approach in that it is provably optimal for linear classifiers. The main insight is a geometric characterization of the decision space that reduces the problem of designing best response oracles to minimizing a quadratic function over a set of convex polytopes.

## 1 INTRODUCTION

In this paper, we study adversarial attacks that induce misclassification when a learner has access to multiple classifiers. One of the most pressing concerns within the field of AI has been the well-demonstrated sensitivity of machine learning algorithms to noise and their general instability. Seminal work by (Szegedy et al., 2014) has shown that adversarial attacks that produce small perturbations can cause data points to be misclassified by state-of-the-art models, including neural networks. In order to evaluate classifiers' robustness and improve their training, adversarial attacks have become a central focus in machine learning and security (Moosavi-Dezfooli et al., 2016; Koh & Liang, 2017; Liu et al., 2017; Nguyen et al., 2015).

Adversarial attacks induce misclassification by perturbing data points past the decision boundary of a particular class. In the case of binary linear classifiers, for example, the optimal perturbation is to push points in the direction perpendicular to the separating hyperplane. For non-linear models there is no general characterization of an optimal perturbation, though attacks designed for linear classifiers tend to generalize well to deep neural networks (Moosavi-Dezfooli et al., 2016).

Since a learner may aggregate decisions using multiple classifiers, a recent line of work has focused on designing attacks on an *ensemble* of different classifiers (Liu et al., 2017; Tramer et al., 2018; Abbasi & Gagné, 2017; He et al., 2017). In particular, this line of work shows that an entire set of state-of-the-art classifiers can be fooled by using an adversarial attack on an ensemble classifier that averages the decisions of the classifiers in that set. Given that attacking an entire set of classifiers is possible, the natural question is then:

*What is the most effective approach to design attacks on a set of multiple classifiers?*

The main challenge when considering attacks on multiple classifiers is that fooling a single model, or even the ensemble classifier (i.e. the model that classifies a data point by averaging individual predictions), provides no guarantees that the learner will fail to classify correctly. Models may have different decision boundaries, and perturbations that affect one may be ineffective on another. Furthermore, a learner can randomize over classifiers and avoid deterministic attacks (see Figure 1).

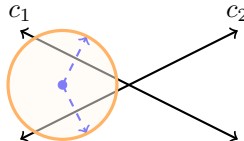

Figure 1: Illustration of why randomization is necessary to compute optimal adversarial attacks. In this example using binary linear classifiers, there is a single point that is initially classified correctly by two classifiers $c_1, c_2$, and a fixed noise budget $\alpha$ in the $\ell_2$ norm. A naive adversary who chooses a noise perturbation deterministically will always fail to trick the learner since she can always select the remaining classifier. An optimal adversarial attack in this scenario consists of randomizing with equal probability amongst both noise vectors.

In this paper, we present a principled approach for attacking a set of classifiers which proves to be highly effective. We show that constructing optimal adversarial attacks against multiple classifiers is equivalent to finding strategies at equilibrium in a zero sum game between a learner and an adversary. It is well known that strategies at equilibrium in a zero sum game can be obtained by applying the celebrated Multiplicative Weights Update framework, given an oracle that computes a best response to a randomized strategy. The main technical challenge we address pertains to the characterization and implementation of such oracles. Our main contributions can be summarized as follows:

- We describe the Noise Synthesis FrameWork (henceforth NSFW) for generating adversarial attacks. This framework reduces the problem of designing optimal adversarial attacks for a general set of classifiers to constructing a best response oracle in a two player, zero sum game between a learner and an adversary;

- We show that NSFW is an effective approach for designing adversarial noise that fools neural networks. In particular, applying projected gradient descent on an appropriately chosen loss function as a proxy for a best response oracle achieves performance that significantly improves upon current state-of-the-art attacks (see results in Figure 2);

- We show that applying projected gradient descent on an appropriately chosen loss function is a well-principled approach. We do so by proving that for linear classifiers such an approach yields an optimal adversarial attack if the equivalent game has a pure Nash equilibrium. This result is shown via a geometric characterization of the decision boundary space which reduces the problem of designing optimal attacks to a convex program;

- If the game does not have a pure Nash equilibrium, there is an algorithm for finding an optimal adversarial attack for linear classifiers whose runtime is exponential in the number of classifiers. We show that finding an optimal strategy in this case is NP-hard.

**Paper organization.** Following a discussion on related work, in Section 2 we formulate the problem of designing optimal adversarial noise and show how it can be modeled as finding strategies at equilibrium in a two player, zero sum game. Afterwards, we discuss our approach for finding such strategies using MWU and proxies for best response oracles. In Section 2.1, we justify our approach by proving guarantees for linear classifiers. Lastly, in Section 3, we present our experiments.

**Additional related work.** The field of adversarial attacks on machine learning classifiers has recently received widespread attention from a variety of perspectives (Carlini & Wagner, 2018; Athalye et al., 2018; Elsayed et al., 2018; Papernot et al., 2016b; Schmidt et al., 2018; Bubeck et al., 2018; Madry et al., 2018). In particular, a significant amount of effort has been devoted to computing adversarial examples that induce misclassification across multiple models (Moosavi-Dezfooli et al., 2017; Szegedy et al., 2014; Moosavi-Dezfooli et al., 2016). There has been compelling evidence which empirically demonstrates the effectiveness of ensembles as way of both generating and defending against adversarial attacks. For example, Tramer et al. (2018) establish the strengths of ensemble training as a defense against adversarial attacks. Conversely, Liu et al. (2017) provide the first set of experiments showing that attacking an ensemble classifier is an effective way of generating adversarial examples that transfer to the underlying models. Relative to their investigation, our work differs in certain key aspects. Rather than analyzing adversarial noise from a security perspective and developing methods for black-box attacks, we approach the problem from a theoretical point of view and introduce a formal characterization of the optimal attack against a set of classifiers. Furthermore, by analyzing noise in the linear setting, we design algorithms for this task that have strong guarantees of performance. Through our experiments, we demonstrate how these algorithms motivate a natural extension for noise in deep learning that achieves state-of-the-art results.

## 2    A FRAMEWORK FOR OPTIMAL ADVERSARIAL ATTACKS

Given a set of point-label pairs $\{(x_i, y_i)\}_{i=1}^m$ where $(x_i, y_i) \in \mathbf{R}^d \times [k]$, a *deterministic adversarial attack* is a totally ordered set of *noise vectors*, $V = (v_1, \ldots, v_m) \in \mathbf{R}^{d \times m}$. We say that $\mathbf{q}$ is an *adversarial attack* if $\mathbf{q}$ is a distribution over sets of noise vectors. An adversarial attack $\mathbf{q}$ is $\alpha$-*bounded* if for all sets $V$ that have non-zero probability under $\mathbf{q}$, each individual noise vector $v_i \in V$ has bounded norm, e.g $||v_i||_p \leq \alpha$. We focus on the case where each vector $v_i$ is bounded to have $\ell_2$ norm less than a fixed value $\alpha$, however, our model can be easily extended to a variety of norms.[1]

For a given classifier $c : \mathbf{R}^d \to [k]$, a realization of the adversarial attack, $V = (v_1, \ldots, v_m)$, induces misclassification on $(x_j, y_j)$ if $c(x_j + v_j) \neq y_j$. Given a finite set of classifiers $\mathcal{C}$ and a data set $S = \{(x_i, y_i)\}_{i=1}^m$ of point-label pairs as above, an *optimal adversarial attack* is a distribution $\mathbf{q}$ over sets of noise vectors that maximizes the minimum 0-1 loss of the classifiers in $\mathcal{C}$:

$$\arg\max_{\mathbf{q}} \min_{c \in \mathcal{C}} \frac{1}{m} \sum_{j \in [m]} \mathbb{E}_{V \sim \mathbf{q}} [\ell_{\text{0-1}}(c, x_j + v_j, y_j)] \tag{1}$$

**Optimal adversarial attacks are equilibrium strategies in a zero sum game.** An equivalent interpretation of the optimization problem described in Equation (1) is that of a *best response* in a two player, zero sum game played between a learner and an adversary. When the learner plays classifier $c \in \mathcal{C}$ and the adversary plays an attack $V$, the payoff to the adversary is $M(c, V) = \frac{1}{m} \sum_{j \in [m]} \ell_{\text{0-1}}(c, x_j + v_j, y_j)$, which is the average 0-1 loss of the learner.[2] The learner and the adversary can choose to play randomized strategies $\mathbf{p}, \mathbf{q}$ over classifiers and noise vectors yielding expected payout $\mathbb{E}_{(c,V) \sim (\mathbf{p},\mathbf{q})} M(c, V)$. The (mixed) equilibrium strategy of the game is the pair of distributions $\mathbf{p}, \mathbf{q}$ that maximize the minimum loss $\max_{\mathbf{q}} \min_{\mathbf{p}} \mathbb{E}_{(c,V) \sim (\mathbf{p},\mathbf{q})} M(c, V)$.

**Computing optimal adversarial attacks via MWU.** As discussed above, the optimization problem of designing optimal adversarial attacks reduces to that of finding strategies at equilibrium in a zero sum game. It is well known that the celebrated Multiplicative Weight Updates algorithm can be used to efficiently compute equilibrium strategies of zero sum games when equipped with a *best response oracle* that finds an optimal set of perturbations for any strategy chosen by the learner:

$$\text{BEST RESPONSE}(\mathbf{p}, \alpha) \stackrel{\text{def}}{=} \arg\max_{V \in \mathbf{R}^{d \times m}} \mathbb{E}_{c \sim \mathbf{p}} [M(c, V)]; \quad \text{s.t } ||v_i||_2 \leq \alpha \; \forall v_i \in V \tag{2}$$

Our framework for generating adversarial noise applies the Multiplicative Weight Updates algorithm as specified in Algorithm 1. The algorithm returns distributions $\mathbf{p}^\star, \mathbf{q}^\star$ that are within $\delta$ of the equilibrium value of the game $\lambda = \min_{\mathbf{p}} \max_{\mathbf{q}} \mathbb{E}_{(c,V) \sim (\mathbf{p},\mathbf{q})}[M(c, V)]$ by using $T \in \mathcal{O}(\frac{\ln n}{\delta^2})$ calls to a best response oracle.[3] In this work, we focus on developing attacks on neural networks and linear models. Yet, our framework is general enough to generate optimal attacks for any domain in which one can approximate a best response. We analyze the convergence of NSFW in Appendix G.

**Approximating a best response.** Given the framework described above, the main challenge is in computing a best response strategy. To do so, at every iteration, as a proxy for a best response, we apply projected gradient descent (PGD) to an appropriately chosen surrogate loss function. In particular, given $\mathcal{C} = \{c_i\}_{i=1}^n$ for every $(x, y) \in \mathbf{R}^d \times [k]$ we aim to solve:

$$\max_v \sum_{i=1}^n \mathbf{p}[i] \ell(c_i, x + v, y) \tag{3}$$

$\ell$ is a loss function that depends on the type of attack (targeted vs. untargeted) and the type of classifiers in $\mathcal{C}$ (linear vs. deep). We introduce a series of alternatives for $\ell$ in the following section.

As we will now show, maximizing the loss of the learner by applying PGD to a weighted sum of loss functions is a well-principled approach to computing best responses as it is guaranteed to converge to the optimal solution in the case where $\mathcal{C}$ is composed of linear classifiers. While there are generally no guarantees for solving non-convex optimization problems of this sort for deep neural networks, in Section 3, we demonstrate the effectiveness of our approach by showing that it experimentally improves upon current state-of-the-art attacks.

---

[1]For example, see Appendix H for extensions to the $\ell_\infty$ norm.

[2]The adversary plays the role of the max player in our two player, zero sum game.

[3]In our experiments in Section 3, we show that the algorithm converges in a far fewer number of iterations.

---

**Algorithm 1** Noise Synthesis FrameWork (NSFW)

---

**Input:** Classifiers $\mathcal{C} = \{c_1, \ldots, c_n\}$, data points $\{(x_i, y_i)\}_{i=1}^{m}$, parameters $\alpha, T$
**initialize** $\mathbf{p}_1 = (\frac{1}{n}, \ldots, \frac{1}{n})$; $\epsilon = \sqrt{\ln |\mathcal{C}|/T}$
**for** $t = 1$ **to** $T$ **do**
   Set $V_t = \text{BEST RESPONSE}(\mathbf{p}_t, \alpha)$
   Set $\mathbf{p}_{t+1}[i] \propto \mathbf{p}_t[i](1-\epsilon)^{M(c_i, V_t)}$ for every $i \in [n]$
**end for**
**Return:** uniform distributions $\mathbf{p}^\star, \mathbf{q}^\star$ over $\mathbf{p}_1, \ldots, \mathbf{p}_T$; $V_1, \ldots, V_T$

---

## 2.1 PROVABLE GUARANTEES FOR COMPUTING OPTIMAL NOISE

The main theoretical insight that leads to provable guarantees for generating adversarial noise is a geometric characterization of the underlying structure of adversarial attacks. Regardless of the type of model, selecting a distribution over classifiers partitions the input space into disjoint regions, each of which is associated with a single loss value for the learner. Given a distribution over classifiers played by the learner, computing a best response strategy for the adversary then reduces to a search problem. In this problem, the search is for points in each region that lie within the noise budget and can be misclassified. The best response is to select the region which induces the maximal loss.

In the case of linear classifiers, the key observation is that the regions are *convex*. As a result, designing optimal adversarial attacks reduces to solving a series of quadratic programs.

**Lemma 1.** *Selecting a distribution* $\mathbf{p}$ *over a set* $\mathcal{C}$ *of* $n$ *linear classifiers, partitions the input space* $\mathbf{R}^d$ *into* $k^n$ *disjoint, convex sets* $T_j$ *such that:*

1. *For each* $T_j$, *there exists a unique label vector* $s_j \in [k]^n$ *such that for all* $x \in T_j$ *and* $c_i \in \mathcal{C}$, $c_i(x) = s_{j,i}$, *where* $s_{j,i}$ *is a particular label in* $[k]$.

2. *There exists a finite set of numbers* $a_1, \ldots a_{k^n}$, *not necessarily all unique, such that* $\sum_{i=1}^{n} \mathbf{p}[i]\ell_{0\text{-}1}(c_i, x, y) = a_j$ *for a fixed* $y$ *and all* $x \in T_j$

3. $\mathbf{R}^d \setminus \bigcup_j T_j$ *is a set of measure zero.*

*Proof Sketch (see full proof in Appendix C).* Each set $T_j$ is defined according to the predictions of the classifiers $c_i \in \mathcal{C}$ on points $x \in T_j$. In particular, each region $T_j$ is associated with a unique label vector $s_j \in [k]^n$ s.t $c_i(x) = s_{j,i}$ for all $c_i \in \mathcal{C}$. Since the prediction of each classifier is the same for all points in a particular region, the loss of the learner $\sum_{i \in [n]} \mathbf{p}[i]\ell_{0\text{-}1}(c_i, x, y)$ is constant over the entire region. Convexity then follows by showing that each $T_j$ is an intersection of hyperplanes. □

This characterization of the underlying geometry now allows us to design best response oracles for linear classifiers via convex optimization. For our analysis, we focus on the case where $\mathcal{C}$ consists of "one-vs-all" classifiers. In the appendix, we show how our results can be generalized to other methods for multilabel classification by reducing these other approaches to the "one-vs-all" case. Given $k$ classes, a "one-vs-all" classifier $c_i$ consists of $k$ linear functions $c_{i,j}(x) = \langle w_{i,j}, x \rangle + b_{i,j}$ where $j \in [k]$. On input $x$, predictions are made according to the rule $c_i(x) = \arg\max_j c_{i,j}(x)$.

**Lemma 2.** *For linear classifiers, implementing a best response oracle reduces to the problem of minimizing a quadratic function over a set of* $k^n$ *convex polytopes.*

*Proof Sketch (see full proof in Appendix C).* The main idea behind this lemma is that given a distribution over classifiers, the loss of the learner can be maximized individually for each point $(x, y) \in S$. Furthermore, by Lemma 1, the loss can assume only finitely many values, each of which is associated with a particular convex region $T_j$ of the input space. Therefore, to compute a best response, we can iterate over all regions and choose the one associated with the highest loss. To find points in each region $T_j$, we can simply minimize the $\ell_2$ norm of a perturbation $v$ such that $x + v \in T_j$, which can be framed as minimizing a quadratic function over a convex set. □

These results give an important characterization, but it also shows that the number of polytopes is exponential in the number of classifiers. To overcome this difficulty, we demonstrate how when there exists a *pure strategy Nash equilibrium* (PSNE), that is a single set of noise vectors $V$ where every vector is bounded by $\alpha$ and $\min_{c_i \in \mathcal{C}} M(c_i, V) = 1$, PGD applied to the *reverse hinge loss*, $\ell_r$, is

guaranteed to converge to a point that achieves this maximum for binary classifiers. More generally, given a label vector $s_j \in [k]^n$ , PGD applied to the *targeted reverse hinge loss*, $\ell_t$, converges to a point within the noise budget that lies within the specified set $T_j$. We define $\ell_r$ and $\ell_t$ as follows:

$$\ell_r(c_i, x, y) \stackrel{\text{def}}{=} \left(y(\langle w_i, x \rangle + b_i)\right)^+; \qquad \ell_t(c_i, x, j) \stackrel{\text{def}}{=} \left(\max_{l \neq j} c_{i,l}(x) - c_{i,j}\right)^+ \qquad (4)$$

The proof follows standard arguments for convergence of convex and $\beta$-smooth functions.

**Theorem 1.** *Given any precision $\epsilon > 0$ and noise budget $\alpha > 0$:*

- *For a finite set of linear binary classifiers $\mathcal{C}$ and a point $(x, y)$, running PGD for $T = 4\alpha/\epsilon$ iterations on the objective $f(v) = \sum_{i=i}^{n} \mathbf{p}[i]\ell_r(c_i, x + v, y)$ converges to a point that is within $\epsilon$ of the pure strategy Nash equilibrium $f(x + v^*)$, if such an equilibrium exists;*

- *For a finite set of linear multilabel classifiers $\mathcal{C}$, given a label vector $s_j \in [k]^n$ and a distribution $\mathbf{p}$ over $\mathcal{C}$, running PGD for $T = 4\alpha/\epsilon$ iterations on the objective $f(v) = \sum_{i=i}^{n} \mathbf{p}[i]\ell_t(c_i, x+v, s_{j,i})$ converges to a point $x+v^{(T)}$ such that $f(x+v^{(T)})-f(x+v^*) \leq \epsilon$ where $x + v^* \in T_j$ and $||v^*||_2 \leq \alpha$, if such a point exists.*

*Proof Sketch.* From the definition of the reverse hinge loss, we see that $\ell_r(c_i, x', y) = 0$ if and only if $\ell_{0\text{-}1}(c_i, x', y) = 1$. Similarly, the targeted loss $\ell_t(c_i, x', j)$ is 0 if and only if $c_i$ predicts $x'$ to have label $j$. For linear classifiers, both of these functions are convex and $\beta$-smooth. Hence PGD converges to a global minimum, which is zero if there exists a pure equilibrium in the game. □

The requirement that there exist a feasible point $x'$ within $T_j$ is not only sufficient, it is also necessary in order to avoid a brute force search. Designing an efficient algorithm to find the region associated with the highest loss is unlikely as the decision version of the problem is NP-hard even for binary linear classifiers. We state the theorem below and defer the proof to the appendix.

**Theorem 2.** *Given a set $\mathcal{C}$ of $n$ binary, linear classifiers, a number $B$, a point $(x, y)$, noise budget $\alpha$, and a distribution $\mathbf{p}$, finding $v$ with $||v||_2 \leq \alpha$ s.t. the loss of the learner is exactly $B$ is NP-complete.*

As we show in the following section, this hardness result does not limit our ability to compute optimal adversarial examples. Most of the problems that have been examined in the context of adversarial noise suppose that the learner has access only to a small number of classifiers (e.g less than 5) (Liu et al., 2017; Dong et al., 2017; Abbasi & Gagné, 2017; Tramer et al., 2018; He et al., 2017). In such cases we can solve the convex program over all regions and find an optimal adversarial attack, even when a pure Nash equilibrium does not exist.

## 3 EXPERIMENTS

We evaluate the performance of NSFW at fooling a set of classifiers by comparing against noise generated by using state-of-the-art attacks against an ensemble classifier. Recent work by Liu et al. (2017) and Tramer et al. (2018), demonstrates how attacking an ensemble of a set of classifiers generates noise that improves upon all previous attempts at fooling multiple classifiers. We test our methods on deep neural networks on MNIST and ImageNet, as well as on linear classifiers where we know that NSFW is guaranteed to converge to the optimal adversarial attack.

### 3.1 EVALUATING NSFW ON DEEP NEURAL NETWORKS

We use the insights derived from our theoretical analysis of linear models to approximate a best response oracle for this new setting. Specifically, at each iteration of NSFW we compute a best response as in Equation (3) by running PGD on a weighted sum of *untargeted reverse hinge losses*, $\ell_{ut}$, introduced in this domain by Carlini & Wagner (2017). Given a network $c_i$, we denote $c_{i,j}(x)$ to be the probability assigned by the model to input $x$ belonging to class $j$ (the $j$th output of the softmax layer of the model).

$$\ell_{ut}(c_i, x, y) \stackrel{\text{def}}{=} \left(c_{i,y}(x) - \max_{j \neq y} c_{i,j}(x)\right)^+ \qquad (5)$$

For MNIST, the set of classifiers $\mathcal{C}$ consists of 5 convolutional neural networks, each with a different architecture, that we train on the full training set of 55k images (see Appendix for details). All classifiers (models) were over 97% accurate on the MNIST test set. For ImageNet, $\mathcal{C}$ consists of the InceptionV3, DenseNet121, ResNet50, VGG16, and Xception models with pre-trained weights

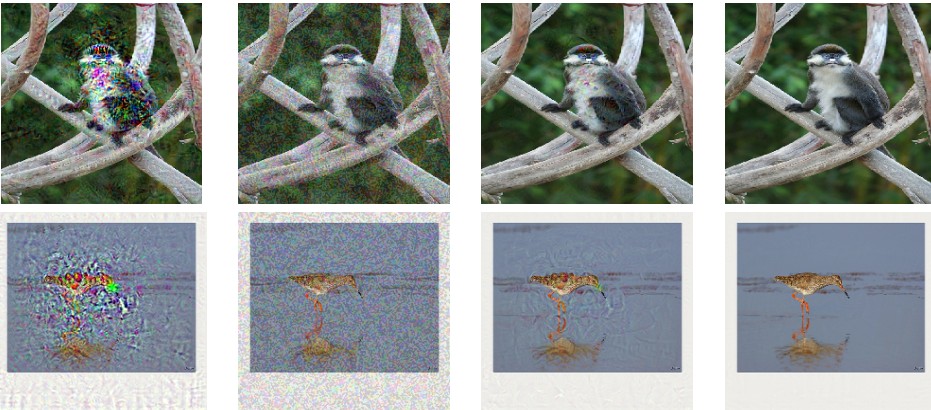

Figure 2: Visual comparison of misclassification using state-of-the-art adversarial attacks. We compare the level of noise necessary to induce similar levels of misclassification by attacking an ensemble classifier using the (from left to right) Fast Gradient Method (FGM), the Madry attack, and the Momentum Iterative Method (MIM) versus applying NSFW (rightmost column) on the same set of classifiers. To induce a maximum of 17% accuracy across all models, we only need to set $\alpha$ to be 300 for NSFW. For the MIM attack on the ensemble we need to set $\alpha = 2000$. For FGM and the Madry attack, the noise budget must be further increased to 8000.

downloaded from the Keras repository (Chollet et al., 2015; He et al., 2016; Simonyan & Zisserman, 2014; Chollet, 2017; Szegedy et al., 2016; Huang et al., 2017).[4]

To evaluate the merits of our approach, we compare our results against attacks on the ensemble composed of $\mathcal{C}$ as suggested by Liu et al. (2017). More specifically, we create an ensemble by averaging the outputs of the softmax layers of the different networks using equal weights. We generate baseline attacks by attacking the ensemble using (1) the Fast Gradient Method by Goodfellow et al. (2014), (2) the Projected Gradient Method by Madry et al. (2018), and (3) the Momentum Iterative Method by Dong et al. (2017) which we download from the Cleverhans library (Papernot et al., 2016a).[5]

We select the noise budget $\alpha$ by comparing against the average $\ell_2$ distortion reported by similar papers in the field. For MNIST, we base ourselves off the values reported by Carlini & Wagner (2017) and choose a noise budget of 3.0. For ImageNet, we compare against Liu et al. (2017). In their paper, they run similar untargeted experiments on ImageNet with 100 images and report a noise budget of 22 when measured as the root mean squared deviation. Converted to the $\ell_2$ norm, this corresponds to $\alpha \geq 8500$.[6] We found this noise budget to be excessive, yielding images comparable to those in the leftmost column in Figure 2. Therefore, we chose $\alpha = 300$ (roughly 3.5% of the total distortion used in Liu et al. (2017)) which ensures that the perturbed images are visually indistinguishable from the originals to the human eye (see rightmost column in Figure 2).

| Noise Algorithm | InceptionV3 | Xception | ResNet50 | DenseNet121 | VGG16 | Mean | Max |
|---|---|---|---|---|---|---|---|
| FGM | 74% | 77% | 60% | 54% | 66% | 66 % | 77 % |
| Madry Attack | 74% | 76% | 58% | 53% | 73% | 67% | 76 % |
| Momentum Iterative Method | 68% | 65% | 34 % | 35% | 49% | 50% | 68% |
| **NSFW** | **17%** | **12.2%** | **5.8%** | **7.2%** | **13.4%** | **11%** | **17%** |

Table 1: Accuracies of ImageNet models under different noise algorithms using a noise budget of 300.0 in the $\ell_2$ norm. Entry $(i, j)$ indicates the accuracy of each model $j$ when evaluated on noise from attack $i$. The last two columns report the mean and max accuracy of the classifiers on a particular attack. We see that NSFW significantly outperforms noise generated by an ensemble classifier for all choices of attack algorithms.

---

[4]Specific details regarding model architectures as well as the code for all our experiments can be found in our repository which will be made public after the review period in order to comply with anonymity guidelines. The test set accuracies of all ImageNet classifiers are displayed on the Keras website.

[5]Momentum Iterative Method won the 2017 NIPS adversarial attacks competition (Kurakin et al., 2018).

[6]In their paper, Liu et al. (2017) define the root mean squared deviation of two points $x, x^\star$ as $\sqrt{\sum (x_i - x_i^\star)^2 / N}$ where $N$ is the dimension of $x$. For ImageNet, our images are of dimension $224 \times 224 \times 3$, while for MNIST they are of size $28 \times 28 \times 1$. For further perspective, if we convert our noise budgets from the $\ell_2$ norm to RMSD, our budgets would correspond to .77 and .11 for ImageNet and MNIST respectively.

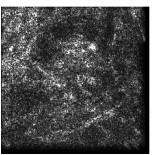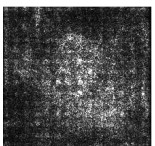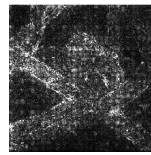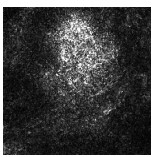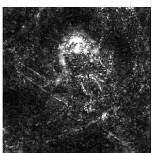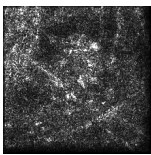

Figure 3: Class saliency map with respect to the image displayed in the top row of Figure 2 for each ImageNet classifier and their ensemble. From left to right: InceptionV3, Xception, ResNet50, DenseNet121, VGG16, and the ensemble classifier of all 5 models.

For our experiments, we ran NSFW for 50 MWU iterations on MNIST models and for 10 iterations on ImageNet classifiers. We use far fewer iterations than the theoretical bound since we found that in practice NSFW converges to the equilibrium solution in only a small number of iterations (see Figure 5 in Appendix A). At each iteration of the MWU we approximate a best response as described in Equation 3 by running PGD using the Adam optimizer (Kingma & Ba, 2014) on a sum of untargeted reverse hinge losses. Specifically, we run the optimizer for 5k iterations with a learning rate of .01. At each iteration, we clip images to lie in the range $[0, 1]$ for MNIST and $[0, 255.0]$ for ImageNet.[7]

Finally, for evaluation, for both MNIST and ImageNet we selected 100 images uniformly at random from the set of images in the test sets that were correctly classified by all models. In Table 1, we report the empirical accuracy of all classifiers in the set $\mathcal{C}$ when evaluated on NSFW as well as on the three baseline attacks. To compare their performance, we highlight the average and maximum accuracies of models in $\mathcal{C}$ when attacked using a particular noise solution.

From Table 1, we see that on ImageNet our algorithm results in solutions that robustly optimize over the entire set of models using only a small amount of noise. The maximum accuracy of any classifier is 17% under NSFW, while the best ensemble attack yields a max accuracy of only 68%. If we wish to generate a similar level of performance from the ensemble baselines, we would need to increase the noise budget to 8000 for FGM and the Madry attack and to 2000 for the Momentum Iterative Method. We present a visual comparison of the different attacks under these noise budgets required to achieve accuracy of 17% in Figure 2. On MNIST, we find similar results. NSFW yields a max accuracy of 22.6% compared to the next best result of 48% generated by the Madry attack on the ensemble. We summarize the results for MNIST in Table 2 presented in Appendix A.

### 3.2 WHY ARE DIRECT ATTACKS ON ENSEMBLE NETWORKS POOR NOISE GENERATORS? ANALYZING DIFFERENCES IN DECISION BOUNDARIES VIA CLASS SALIENCY MAPS

As seen in the previous section, noise generated by directly attacking an ensemble of classifiers significantly underperforms NSFW at robustly fooling the underlying models. In this section, we aim to understand this phenomenon by analyzing how the decision boundary of the ensemble model compares to that of the different networks. In particular, we visualize the class boundaries of convolutional neural networks using the algorithm proposed by Simonyan et al. (2013) for generating *saliency maps*.[8] The class saliency map indicates which features (pixels) are most relevant in classifying an image to have a particular label. [9] Therefore, they serve as one way of understanding the decision boundary of a particular model by highlighting which dimensions carry the highest weight.

In Figure 3, we see that the class saliency maps for individual models exhibit significant diversity. The ensemble of all 5 classifiers appears to contain information from all models, however, certain regions that are of central importance for individual models are relatively less prominent in the ensemble saliency map. Compared to our approach which calculates individual gradients for classifiers in $\mathcal{C}$, creating an ensemble classifier obfuscates key information regarding the decision boundary of individual models. We make this discussion rigorous by analyzing the linear case in Appendix B.

---

[7]Further discussion regarding experiment setup and hyperparameters may be found in Appendix A.

[8]We make use of the implementation found at https://github.com/experiencor/deep-viz-keras.

[9]If we let $c_{i,j}(x)$ be the $j$th output of the softmax layer of network $c_i$, they define the image-specific class saliency as the derivative $\partial c_{i,j}(x)/\partial x$. In the case of multichannel images, the value per pixel is defined as the maximum across all channels so as to yield a single grayscale image.

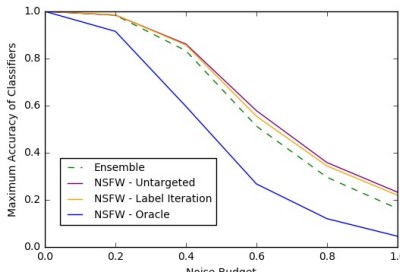 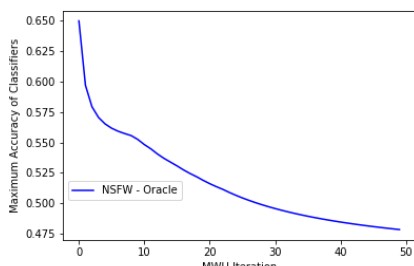

Figure 4: Results of running NSFW on linear models. On the left, we demonstrate the results of running NSFW on linear multiclass models using different noise functions and varying the noise budget $\alpha$. NSFW-Oracle corresponds to running Algorithm 1 using the best response oracle described in Lemma 2. Similarly, NSFW-Untargeted shows the results of running NSFW and applying PGD to a weighted sum of untargeted losses as in Equation (3). The label iteration method is described below. Lastly, the ensemble attack corresponds to the optimal noise on an equal weights ensemble of models in $\mathcal{C}$. On the right, we illustrate the convergence of NSFW on linear binary classifiers with maximally different decision boundaries to compare against the convergence rate observed for neural nets in Figure 5 and better understand when weight adaptivity is necessary.

### 3 .3 EXPERIMENTS ON LINEAR CLASSIFIERS

In addition to evaluating our approach on neural networks, we performed experiments with linear classifiers. Since we have a precise characterization of the optimal attack on a set of linear classifiers, we can rigorously analyze the performance of different methods in comparison to the optimum.

We train two sets of 5 linear SVM classifiers on MNIST, one for binary classification (digits 4 and 9) and another for multiclass (first 4 classes, MNIST 0-3). To ensure a diversity of models, we randomly zero out up to 75% of the dimensions of the training set for each classifier. Hence, each model operates on a random subset of features. All models achieve test accuracies of above 90%. For our experiments, we select 1k points from each dataset that are correctly classified by all models.

In order to better compare across different best response proxies, we further extend NSFW by incorporating the *label iteration method* as another heuristic to generate untargeted noise. Given a point $(x, y)$, the iterative label method attempts to calculate a best response by running PGD on the targeted reverse hinge loss for every label $j \in [k] \setminus \{y\}$ and choosing the attack associated with the minimal loss. Compared to the untargeted reverse hinge loss, it has the benefit of being convex.

As for deep learning classifiers, we compare our results to the noise generated by the optimal attack on an ensemble of models in $\mathcal{C}$. Since the class of linear classifiers is convex, creating an equal weights ensemble by averaging the weight vectors results in just another linear classifier. We can compute the optimal attack by running the best response oracle described in Section 2 .1 for the special case where $\mathcal{C}$ consists of a single model and then scaling the noise to have norm equal to $\alpha$.

As seen in the leftmost plot in Figure 4, even for linear models there is a significant difference between the optimal attack and other approaches. Specifically, we observe an empirical gap between NSFW equipped with the best response oracle as described in Lemma 2 vs. NSFW with proxy best response oracles, e.g. the oracle that runs PGD on appropriately chosen loss functions.[10] This difference in performance is consistent across a variety of noise budgets. Our main takeaway is that in theory and in practice, there is a significant benefit in applying appropriately designed best response oracles. Lastly, on the right in Figure 4, we illustrate how the adaptivity of MWU is in general necessary to compute optimal attacks. While for most cases, NSFW converges to the equilibrium solution almost immediately, if the set of classifiers is sufficiently diverse, running NSFW for a larger number of rounds drastically boosts the quality of the attack. (See Appendix A for details.)

### 4 CONCLUSION

Designing adversarial attacks when a learner has access to multiple classifiers is a non-trivial problem. In this paper we introduced NSFW which is a principled approach that is provably optimal on linear classifiers and empirically effective on neural networks. The main technical crux is in designing best response oracles which we achieve through a geometrical characterization of the optimization landscape. We believe NSFW can generalize to domains beyond those in this paper.

---

[10]We present a similar figure for binary classifiers in the appendix.

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

# Appendix: Supplementary Material

## A  ADDITIONAL EXPERIMENTS AND DETAILS ON EXPERIMENTAL SETUP

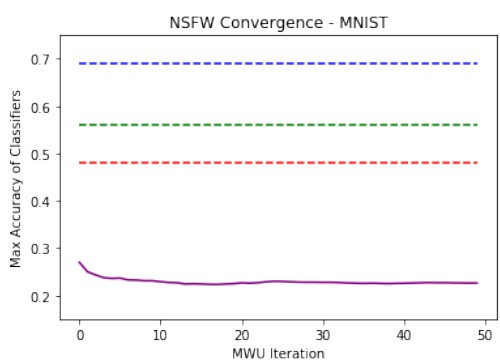 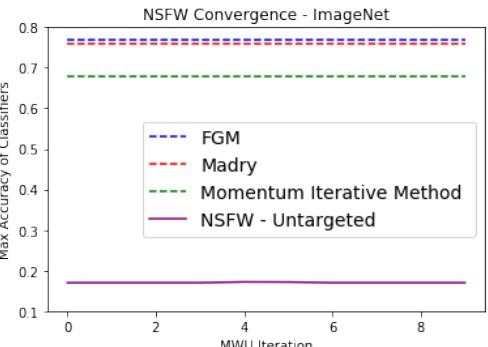

Figure 5: Fast convergence of NSFW on MNIST and ImageNet deep learning models. NSFW-Untargeted corresponds to running NSFW and applying PGD on a sum of untargeted reverese hinge losses as described in Section 3 .1. The dotted lines correspond to running the indicated attack on the ensemble of models in $\mathcal{C}$. For both datasets, we find that NSFW converges almost immediately to the equilibrium noise solution. These misclassification results can also be examined in Tables 1 and 2.

We now discuss further details regarding the setup of the experiments presented in Section 3 . In the case of deep learning, we set hyperparameters for of all the baseline attacks (Fast Gradient Method, Madry attack, and the Momentum Iterative Method) by analyzing the values reported in the original papers. When running the Projected Gradient Method by Madry et al., given a noise budget $\alpha$, we run the algorithm for 40 iterations with a step size of $\alpha/40 \times 1.25$ so as to mimic the setup of the authors. In the case of the Momentum Iterative Method, we run the attack for 5 iterations with a decay factor $\mu = 1.0$ and a step size of $\alpha/5$ as specified in Dong et al. (2017). FGM has no hyperparameters other than the noise budget. For all methods, we clip solutions to lie within the desired pixel range and noise budget.

When comparing different algorithms to compute best responses for linear multiclass classifiers as described in Section 3 .3, we run the NSFW algorithm with $\alpha = .2k$ for $k \in [5]$. In the case of binary classifiers (Figure 6), we find that the margins are smaller, and hence run NSFW with $\alpha = .05 + .1k$ for $k \in [5]$. For each value of $\alpha$ and choice of noise function, we run NSFW for 50 iterations. The one exception is that, for the multiclass experiments with $\alpha$ equal to .2 or .4, we ran the best response oracle for only 20 iterations due to computational constraints. When optimizing the loss of the learner through gradient descent (e.g when using PGD on appropriately chosen loses), we set the number of iterations to 3k and the learning rate to .01.

We set up the weight adaptivity experiment described at the end of Section 3 .3 (rightmost plot of Figure 4) as follows. We train 5 linear binary SVM classifiers on our binary version of the MNIST dataset. For each classifier, we zero out 80% of the input dimensions so that each model has nonzero weights for a strictly different subset of features, thereby ensuring maximum diversity in the decision

| Noise Algorithm | Model 1 | Model 2 | Model 3 | Model 4 | Model 5 | Mean | Max |
|---|---|---|---|---|---|---|---|
| FGM | 59% | 63% | 60% | 69% | 59% | 62% | 69% |
| Madry Attack | 48% | 43% | 33% | 36% | 40% | 40% | 48% |
| Momentum Iterative Method | 45% | 50% | 44% | 56% | 39% | 46.8% | 56% |
| **NSFW** | **20.2%** | **16.2%** | **11.6%** | **22.6%** | **14.6%** | **17%** | **22.6%** |

Table 2: Classification accuracies for deep learning MNIST models under different noise algorithms. As in the ImageNet case, we find that the NSFW algorithm improves upon the performance of state-of-the-art attacks and robustly optimizes over the entire set of classifiers. Moreover, we find that, for all attacks, there is a significant difference between the average and maximum accuracy of classifiers in $\mathcal{C}$, further highlighting the need to design noise algorithms that are guaranteed to inhibit the performance of the best possible model.

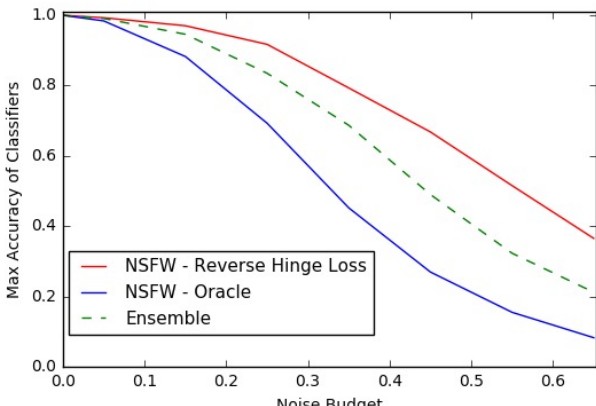

Figure 6: Results of running NSFW for linear binary classifiers and varying the noise budget. As seen in the case of multiclass classifiers in Figure 4 NSFW equipped with the best response outperforms other approaches at generating noise for linear models. Furthermore, we see there is a performance gap between gradient based approaches and the theoretically optimal one that leverages convex programming.

boundaries across models. In order to generate noise, we select 500 points uniformly at random from the test set that were correctly classified by all modes. We then run NSFW equipped with the best response oracle described in Lemma 2 for 50 iterations with a noise budget of .4.

## B  WHY ATTACKING AN ENSEMBLE IS NOT OPTIMAL

In this section, we provide a brief theoretical justification as to why methods designed to attack an ensemble constructed by averaging individual models in a set of classifiers (e.g. Liu et al. (2017); Tramer et al. (2018); He et al. (2017); Abbasi & Gagné (2017)) do not yield optimal adversarial attacks against a learner who selects classifiers from this set.

Attacks on ensemble classifiers, as seen in Liu et al. (2017), typically consist of applying gradient based optimization to an ensemble model $E(\mathcal{C}, \mathbf{p})$ made up of classifiers $\mathcal{C}$ and ensemble weights $\mathbf{p}$. For concreteness, consider the simplest case where $\mathcal{C}$ is composed of linear binary classifiers. To find adversarial examples, we run gradient descent on a loss function such as the reverse hinge loss that is 0 if and only if the perturbed example $x' = x + v$ with true label $y$ is misclassified by $c_i$.

Assuming $x'$ is not yet misclassified by the ensemble, running SGD on the ensemble classifier with the reverse hinge loss function results in a gradient update step of $\nabla \ell_r(E(\mathcal{C}, \mathbf{p}), x', y) = \sum_i \mathbf{p}[i] w_i$. This is undesirable for two main reasons:

- First, the ensemble obscures valuable information about the underlying objective. If $x'$ is misclassified by a particular model $c_i$ but not the ensemble, $c_i$ still contributes $\mathbf{p}[i] w_i$ to the resulting gradient and biases exploration away from promising regions of the search space;

- Second, fooling the ensemble does not guarantee that the noise will transfer across the underlying models. Assuming the true label $y$ is -1 and that $x'$ is correctly classified by all models, $\ell_r(E(\mathcal{C}, \mathbf{p}), x', y) = 0$ if and only if there exists a subset of classifiers $\mathcal{T} \subseteq \mathcal{C}$ such that $\sum_{c_t \in \mathcal{T}} \mathbf{p}[t](\langle w_t, x' \rangle + b_t) > 0$, $\mathbf{p}[j](\sum_{c_j \in \mathcal{C} \setminus \mathcal{T}} \langle w_j, x' \rangle + b_j) < 0$ and $\sum_{c_t \in \mathcal{T}} \mathbf{p}[t](\langle w_t, x' \rangle + b_t) > |\mathbf{p}[j](\sum_{c_j \in \mathcal{C} \setminus \mathcal{T}} \langle w_j, x' \rangle + b_j)|$. Hence, the strength of an ensemble classifier is only as good as its weakest weighted majority.

## C   PROOF OF GEOMETRIC CHARACTERIZATION

**Lemma 1.** *Selecting a distribution* $\mathbf{p}$ *over a set* $\mathcal{C}$ *of* $n$ *linear classifiers, partitions the input space* $\mathbf{R}^d$ *into* $k^n$ *disjoint, convex sets* $T_j$ *such that:*

1. *For each* $T_j$, *there exists a unique label vector* $s_j \in [k]^n$ *such that for all* $x \in T_j$ *and* $c_i \in \mathcal{C}$, $c_i(x) = s_{j,i}$, *where* $s_{j,i}$ *is a particular label in* $[k]$.

2. *There exists a finite set of numbers* $a_1, \ldots a_{k^n}$, *not necessarily all unique, such that* $\sum_{i=1}^{n} \mathbf{p}[i]\ell_{0\text{-}1}(c_i, x, y) = a_j$ *for a fixed* $y$ *and all* $x \in T_j$

3. $\mathbf{R}^d \setminus \bigcup_j T_j$ *is a set of measure zero.*

*Proof.* Given a label vector $s_j$, we define each $T_j$ as the set of points $x$ where $c_i(x) = s_{j,i}$ for all $i \in [n]$. This establishes a bijection between the elements of $[k]^n$ and the sets $T_j$. All the $T_j$ are pairwise disjoint since their corresponding label vectors in $[k]^n$ must differ in at least one index and by construction each classifier can only predict a single label for $x \in T_j$.

To see that these sets are convex, consider points $x_1, x_2 \in T_j$ and an arbitrary classifier $c_i \in \mathcal{C}$ s.t. $c_i(x) = z$ for all $x \in T_j$. If we let $x' = \gamma x_1 + (1 - \gamma)x_2$ where $\gamma \in [0,1]$ then the following holds for all $j \in [k]$ where $j \neq z$:

$$
\begin{aligned}
c_{i,z}(x') &= \langle w_{i,z}, \gamma x_1 + (1\text{-}\gamma)x_2 \rangle + b_{i,z} \\
&= \gamma \langle w_{i,z}, x_1 \rangle + \gamma b_{i,z} + (1\text{-}\gamma)\langle w_{i,z}, x_2 \rangle + (1\text{-}\gamma)b_{i,z} \\
&> \gamma \langle w_{i,j}, x_1 \rangle + \gamma b_{i,j} + (1\text{-}\gamma)\langle w_{i,j}, x_2 \rangle + (1\text{-}\gamma)b_{i,j} \\
&= c_{i,j}(x')
\end{aligned}
$$

Furthermore, for each $T_j$, there exists a number $a_j \in \mathbf{R}_{\geq 0}$ such that the expected loss of the learner $\sum_i \mathbf{p}[i] \cdot l(c_i, x, y)$ equals $a_j$ for all $x \in T_j$. Since the distribution $\mathbf{p}$ is fixed, the loss of the learner is uniquely determined by the correctness of the predictions of all the individual classifiers $c_i$. Since these are the same for all points in $T_j$, the loss of the learner remains constant.

Lastly, the set $\mathbf{R}^d \setminus \bigcup_i T_i$ is equal to the set of points $x$ where there are ties for the maximum valued classifier. This set is a subset of the set of points $\mathcal{K}$ that lie at the intersection of two hyperplanes:

$$
\mathbf{R}^d \setminus \bigcup_i T_i \subset \{x | \exists\ c_{i,k}, c_{j,l} \text{ s.t } c_{i,k}(x) = c_{j,l}(x)\} \tag{6}
$$

Finally, we argue that $\mathcal{K}$ has measure zero. For all $\varepsilon > 0, x \in \mathcal{K}$, there exists an $x'$ such that $||x - x'||_2 < \varepsilon$ and $x' \notin \mathcal{K}$ since the intersection of two distinct hyperplanes is of dimension two less than the overall space. Therefore, $\mathbf{R}^d \setminus \bigcup_i T_i$ must also have measure zero. □

**Lemma 2.** *For linear classifiers, implementing a best response oracle reduces to the problem of minimizing a quadratic function over a set of* $k^n$ *convex polytopes.*

*Proof.* We outline the proof as follows. Given a distribution $\mathbf{p}$ over $\mathcal{C}$, the loss of the learner $\frac{1}{m}\sum_{t=1}^{m}\sum_{i=i}^{n}\mathbf{p}[i]\ell_{0\text{-}1}(c_i, x_t + v_t, y_t)$ can be optimized individually for each $v_t$ since the terms in the sum are independent from one another. We leverage our results from Lemma 1 to demonstrate how we can frame the problem of finding the minimum perturbation $v_j$ such that $x + v_j \in T_j$ as the minimization of a convex function over a convex set. Since the loss of the learner is constant for points that lie in a particular set $T_j$, we can find the optimal solution by iterating over all sets $T_j$ and selecting the perturbation with $\ell_2$ norm less than $\alpha$ that is associated with the highest loss. The best response oracle then follows by repeating the same process for each point $(x, y)$.

Given a point $(x, y)$ solving for the minimal perturbation $v$ such that $x + v \in T_j$ can be expressed as the minimization of a quadratic function subject to $n(k - 1)$ linear inequalities.

$$\min_{v \in \mathbf{R}^d} \quad ||v||_2^2$$
$$\text{subject to} \quad c_1(x+v) = s_{j,1} \tag{7}$$
$$\ldots$$
$$c_n(x+v) = s_{j,n}$$

Each constraint in (7) can be expressed as $k-1$ linear inequalities. For a particular $z \in [k], c_i \in \mathcal{C}$ we write $c_i(x+v) = z$ as $c_{i,z}(x+v) > c_{i,l}(x+v)$ for all $l \neq z$. Lastly, squaring the norm of the vector is a monotonic transformation and hence does not alter the underlying minimum. □

## D    BEYOND "ONE-VS-ALL" LINEAR CLASSIFICATION

Here we extend the results from our analysis of linear classifiers to other methods for multilabel classification. In particular, we show that any "all-pairs" or multivector model can be converted to an equivalent "one-vs-all" classifier and hence all of our results also apply to these other approaches.

**All-Pairs.** In the "all-pairs" approach, each linear classifier $c$ consists of $\binom{k}{2}$ linear predictors $c_{i,j}$ trained to predict between labels $i, j \in [k]$. As per convention, we let $c_{i,j}(x) = -c_{j,i}(x)$. Labels are chosen according to the rule:

$$c(x) = \arg\max_{i \in [k]} \sum_{j \neq i} c_{i,j}(x) \tag{8}$$

Given an "all-pairs" classifier $c$, we show how it can be transformed into a "one-vs-all" classifier $c'$ such that $c(x) = c'(x)$ for all $x \in \mathbf{R}^d$.

$$c(x) = \arg\max_{i \in [k]} \sum_{j \neq i} c_{i,j}(x)$$
$$= \arg\max_{i \in [k]} \sum_{j \neq i} \langle w_{i,j}, x \rangle + b_{i,j}$$
$$= \arg\max_{i \in [k]} \langle w_i', x \rangle + b_i'$$
$$= \arg\max_{i \in [k]} \sum_{j \neq i} c_i'(x) = c'(x)$$

**Multivector.** Lastly, we extend our results to multilabel classification done via class-sensitive feature mappings and the multivector construction by again reducing to the "one-vs-all" case. Given a function $\Psi : \mathbf{R}^d \times [k] \to \mathbf{R}^n$, labels are predicted according to the rule:

$$c(x) = \arg\max_{y \in [k]} \langle w, \Psi(x, y) \rangle \tag{9}$$

While there are several choices for the $\Psi$, we focus on the most common, the multivector construction:

$$\Psi(x, y) = [\underbrace{0, \ldots, 0}_{\in \mathbf{R}^{(y-1)(d+1)}}, \underbrace{x_1, \ldots, x_n, 1}_{\in \mathbf{R}^{d+1}}, \underbrace{0, \ldots, 0}_{\in \mathbf{R}^{(k-y)(d+1)}}] \tag{10}$$

$$w = [w_1, \ldots, w_k] \text{ where } w_i \in \mathbf{R}^{d+1} \, \forall i \tag{11}$$

This in effect ensures that (9) becomes equivalent to that of the "one-vs-all" approach:

$$c(x) = \arg\max_{i \in [k]} \langle w_i, x \rangle \tag{12}$$

# E  CONVERGENCE ANALYSIS OF PROJECTED GRADIENT DESCENT

**Theorem 1.** *Given any precision $\epsilon > 0$ and noise budget $\alpha > 0$:*

- *For a finite set of linear binary classifiers $\mathcal{C}$ and a point $(x, y)$, running PGD for $T = 4\alpha/\epsilon$ iterations on the objective $f(v) = \sum_{i=i}^{n} \mathbf{p}[i]\ell_r(c_i, x + v, y)$ converges to a point that is within $\epsilon$ of the pure strategy Nash equilibrium $f(x + v^*)$, if such an equilibrium exists;*

- *For a finite set of linear multilabel classifiers $\mathcal{C}$, given a label vector $s_j \in [k]^n$ and a distribution $\mathbf{p}$ over $\mathcal{C}$, running PGD for $T = 4\alpha/\epsilon$ iterations on the objective $f(v) = \sum_{i=i}^{n} \mathbf{p}[i]\ell_t(c_i, x+v, s_{j,i})$ converges to a point $x+v^{(T)}$ such that $f(x+v^{(T)})-f(x+v^*) \leq \epsilon$ where $x + v^* \in T_j$ and $||v^*||_2 \leq \alpha$, if such a point exists.*

*Proof.* We know that if a function $f$ is convex and $\beta$-smooth, then running projected gradient descent over a convex set, results in the following rate of convergence, where $v^\star$ is the optimal solution and $v^{(1)}$ is the initial starting point (See Theorem 3.7 in Bubeck (2015)).

$$f(v^{(T)}) - f(v^\star) \leq \frac{3\beta||v^{(T)} - v^\star||_2 + f(v^{(1)}) - f(v^\star)}{T} \tag{13}$$

Given $n$ classifiers, the objective is $\sum_{i=1}^{n} \mathbf{p}[i]||w_i||_2$ smooth:

$$f(v) = \sum_{i=i}^{n} \mathbf{p}[i]\ell_r(c_i, x + v, y)$$

$$= \sum_{i=i}^{n} \mathbf{p}[i]\big(y(\langle w_i, x + v \rangle + b_i)\big)^+$$

$$||\nabla f(v)||_2 \leq \sum_{i=1}^{n} \mathbf{p}[i]||w_i||_2$$

Furthermore, since $v*$ is a pure strategy Nash equilibrium, $f(v^\star) = 0$ and the maximum difference between $f(v) - f(v^\star)$, for any $v$, is bounded by:

$$f(v) - f(v^\star) \leq \sum_{i=1}^{n} \mathbf{p}[i]\big(y(\langle w_i, x + v \rangle + b_i)\big)^+ - 0$$

$$= \sum_{i=1}^{n} \mathbf{p}[i]\big(y\langle w_i, v - v^\star \rangle + y(\langle w_i, x + v^\star \rangle + b_i)\big)^+ \quad \text{(Convexity of Noise Budget)}$$

$$\leq \sum_{i=1}^{n} \mathbf{p}[i]\big(y\langle w_i, v' \rangle)\big)^+ \quad y(\langle w_i, x + v^\star \rangle + b_i) < 0$$

$$\leq \sum_{i=1}^{n} \mathbf{p}[i]||w_i||_2||v'||_2 \quad \text{(Cauchy-Schwartz)}$$

$$\leq \alpha \sum_{i=1}^{n} \mathbf{p}[i]||w_i||_2 \quad \text{(Noise Budget)}$$

Since $||v^{(T)} - v^\star||_2 \leq \alpha$, we have that:

$$f(v^{(T)}) - f(v^\star) \leq \frac{4\alpha \sum_{i=1}^{n} \mathbf{p}[i]||w_i||_2}{T} \tag{14}$$

Lastly, we can normalize all the $w_i$ such that $||w_i||_2 = 1$ without changing the predictions of the $c_i$ and arrive at our desired result.

For the multiclass case, we have that:

$$f(v) = \sum_{i=1}^{n} \mathbf{p}[i] \ell_t(c_i, x + v, s_{j,i})$$

$$= \sum_{i=1}^{n} \mathbf{p}[i] \big( \max_{l \neq s_{j,i}} \langle w_{i,l}, x + v \rangle + b_{i,l} - (\langle w_{i,s_{j,i}}, x + v \rangle + b_{i,s_{j,i}}) \big)^+$$

A similar bound follows by using the logic for the binary case. If we let $v' = v - v^\star$, we know that by our initial assumption, $\max_{l \neq s_{j,i}} \langle w_{i,l}, x + v^\star \rangle + b_{i,l} - \langle w_{i,s_{j,i}}, x + v^\star \rangle - b_{i,s_{j,i}} < 0$. We use this fact in the second line of the following argument.

$$f(v) - f(v*) \leq \sum_{i=1}^{n} \mathbf{p}[i] \big( \max_{l \neq s_{j,i}} \langle w_{i,l}, x + v \rangle + b_{i,l} - (\langle w_{i,s_{j,i}}, x + v \rangle + b_{i,s_{j,i}}) \big)^+ - 0$$

$$\leq \sum_{i=1}^{n} \mathbf{p}[i] \big( \max_{l \neq s_{j,i}} \langle w_{i,l}, v' \rangle - \langle w_{i,l}, v' \rangle \big)$$

$$\leq \alpha \sum_{i=1}^{n} \mathbf{p}[i] \max_{l \neq s_{j,i}} ||w_{i,l}||_2 \qquad \text{(Cauchy-Schwartz, Noise Budget)}$$

Using the fact that all weight vectors $w_{i,j}$ can be transformed to have $\ell_2$ norm equal to 1, we have that $f(v) - f(v*) \leq \alpha \sum_{i=1}^{n} \mathbf{p}[i]$. Lastly, we can check that $\ell_t$ is $\beta$-smooth with $\beta = \alpha \sum_{i=1}^{n} \mathbf{p}[i]$, which yields the same bound as in the binary case. $\qquad \square$

## F    BEST RESPONSE ORACLE IS NP-COMPLETE

**Theorem 2.** *Given a set $\mathcal{C}$ of $n$ binary, linear classifiers, a number $B$, a point $(x, y)$, noise budget $\alpha$, and a distribution $\mathbf{p}$, finding $v$ with $||v||_2 \leq \alpha$ s.t. the loss of the learner is exactly $B$ is NP-complete.*

*Proof.* We can certainly verify in polynomial time that a vector $v$ induces a loss of $B$ simply by calculating the 0-1 loss of each classifier. Therefore the problem is in NP.

To show hardness, we reduce from Subset Sum. Given $n$ numbers $p_1, \ldots p_n$ and a target number $B$,[11] we determine our input space to be $\mathbf{R}^n$, the point $x$ to be the origin, the label $y = -1$, and the noise budget $\alpha = 1$. Next, we create $n$ binary classifiers of the form $c_i(x) = \langle e_i, x \rangle$ where $e_i$ is the $i$th standard basis vector. We let $p_i$ be the probability with which the learner selects classifier $c_i$.[12]

We claim that there is a subset that sums to $B$ if and only if there exists a region $T_j \subset \mathbf{R}^n$ on which the learner achieves loss $B$. Given the parameters of the reduction, the loss of the learner is determined by the sum of the probability weights of classifiers $c_i$ such that $c_i(x + v) = +1$ for points $x + v \in T_j$. If we again identify sets $T_j$ with sign vectors $s_j \in \{\pm 1\}^n$ as per Lemma 2, there is a bijection between the sets $T_j$ and the power set of $\{p_1, \ldots, p_n\}$. A number $p_i$ is in a subset $U_j$ if the $i$th entry of $s_j$ is equal to $+1$.

Lastly, we can check that there are feasible points within each set $T_j$ and hence that all subsets within the original Subset Sum instance are valid. Each $T_j$ simply corresponds to a quadrant of $\mathbf{R}^n$. For any $\varepsilon > 0$ and for any $T_j$, there exists a $v_j$ with $\ell_2$ norm less than $\varepsilon$ such that $x + v_j \in T_j$. Therefore, there is a subset $U_j$ that sums to $B$ if and only if there is a region $T_j$ in which the learner achieves loss $B$. $\qquad \square$

---

[11]Without loss of generality, we can assume that instances of Subset Sum only have values in the range $[0, 1]$. We can reduce from the more general case by simply normalizing inputs to lie in this range.

[12]We can again normalize values so that they form a valid probability distribution.

## G    CONVERGENCE ANALYSIS OF THE NSFW ALGORITHM

To facilitate our analysis, we overload our notation of the payoff function $M(c, V)$ described in Section 2 to accept distributions $\mathbf{p}, \mathbf{q}$ as input. We write $M(\mathbf{p}, \mathbf{q})$ to indicate $\mathbb{E}_{(c,V) \sim (\mathbf{p}, \mathbf{q})} M(c, V)$.

**Theorem 3.** *Given an error parameter $\delta$, after $\mathcal{O}(\frac{\ln n}{\delta^2})$ iterations, Algorithm 1 returns distributions $\mathbf{p}^\star$, $\mathbf{q}^\star$ such that:*

$$\min_{c_i \in \mathcal{C}} M(c_i, \mathbf{q}^\star) \geq \lambda - \delta$$
$$\max_{V} M(\mathbf{p}^\star, V) \leq \lambda + \delta$$

*where $\lambda = \min_{\mathbf{q}} \max_{\mathbf{p}} M(\mathbf{p}, \mathbf{q})$ is the equilibrium value of the game .*

*Proof.* The following analysis draws heavily upon the work of Freund & Schapire (1997), yet the precise treatment follows that of Kale (2007).

By guarantees of the Multiplicative Weights algorithm, we have that for any distribution $\mathbf{p}$ over $[n]$ with losses in $[0, 1]$ the following relationship holds:

$$\sum_{t=1}^{T} M(\mathbf{p}^{(t)}, V^{(t)}) \leq (1 + \varepsilon) \sum_{i=1}^{T} M(\mathbf{p}, V^{(t)}) + \frac{\ln n}{\varepsilon}$$

If we divide by $T$, and note that $M(\mathbf{p}, V) \leq 1$, and $M(\mathbf{p}^{(t)}, V^{(t)}) \geq \lambda$ for all $t$ (due to oracle guarantees), we have that for any distribution $\mathbf{p}$:

$$\lambda^\star \leq \frac{1}{T} \sum_{i=1}^{T} M(\mathbf{p}^{(t)}, V^{(t)})$$
$$\leq \frac{1}{T} \sum_{i=1}^{T} M(\mathbf{p}, V^{(t)}) + \varepsilon + \frac{\ln n}{\varepsilon T}$$

If we let $\tilde{\mathbf{p}}$ be the optimal strategy for the min player, then $M(\tilde{\mathbf{p}}, V) \leq \lambda^\star$ for any $V$. Next, if we set $\epsilon = \frac{\delta}{2}$ and $T = \lceil \frac{4 \ln n}{\delta^2} \rceil$ we get that:

$$\lambda^\star \leq \frac{1}{T} \sum_{i=1}^{T} M(\mathbf{p}^{(t)}, V^{(t)})$$
$$\leq \frac{1}{T} \sum_{i=1}^{T} M(\tilde{\mathbf{p}}, V^{(t)}) + \delta$$
$$\leq \lambda^\star + \delta$$

Therefore $\mathbf{p}^\star$, the uniform distribution over $\mathbf{p}^{(1)}, \ldots, \mathbf{p}^{(T)}$ is an approximately optimal solution for the learner.

For the adversary, we know from the previous equations that the following holds for any strategy $\mathbf{p}$ played by the learner:

$$\lambda^\star \leq \frac{1}{T} \sum_{i=1}^{T} M(\mathbf{p}^{(t)}, V^{(t)}) \leq \frac{1}{T} \sum_{i=1}^{T} M(\mathbf{p}, V^{(t)}) + \delta$$

If we set $\mathbf{q}^\star$ to be the distribution that assigns weight $\frac{|\{t:V^{(t)}=V\}|}{T}$ to the particular set of noise vectors $V$ then we have that for any distribution $\mathbf{p}$:

$$\lambda^\star \leq \frac{1}{T} \sum_{i=1}^{T} M(\mathbf{p}^{(t)}, V^{(t)}) \leq M(\mathbf{p}, \mathbf{q}^\star) + \delta$$

Hence $\mathbf{q}^\star$ is an approximately optimal strategy for the adversary:

$$\lambda^\star - \delta \leq M(\mathbf{p}, \mathbf{q}^\star)$$

$\square$

## H    EXTENSIONS TO $\ell_\infty$ NORM

While we focus on the $\ell_2$ norm as the main metric with which to gauge the magnitude of adversarial noise, our results can be readily extended to function with the $\ell_\infty$ norm. We begin by illustrating how to extend the best response oracle to use the $\ell_\infty$ norm:

**Lemma 3.** *For linear classifiers, implementing a best response oracle under the $\ell_\infty$ norm reduces to the problem of finding feasible points in a series of $k^n$ convex polytopes.*

*Proof.* The proof is identical to that of Lemma 2. The only change we need to make is to slightly modify the optimization problem outlined in (7) to the following format. Given a label vector $s_j$ and a point $(x, y)$ we solve for:

$$
\begin{aligned}
\min_{v \in \mathbf{R}^d} \quad & 0 \\
\text{subject to} \quad & c_i(x + v) = s_{j,i} \quad \forall i \in [k] \\
& v_i \leq \alpha \quad\quad\quad\ \forall i \in [d]
\end{aligned}
\tag{15}
$$

$\square$

To generalize gradient methods, we can alter the projection step of gradient descent to constrain noise to the $\ell_\infty$ ball. The solution space remains convex and hence our theoretical guarantees still hold.

