# OpenReview forum: "Optimal Attacks against Multiple Classifiers"
_ICLR.cc/2019/Conference_

### Official Review · AnonReviewer3 · 2018-11-02
**Application to multiplicative weight update for constructing adversarial noise**

**Rating:** 6
**Confidence:** 3

**Review:**

Summary:
The paper considers finding the most adversarial random noise given multiple classifiers. They formulate the problem as the standard min-max game and apply the multiplicative weight updates. The technical contribution is to clarify the computational complexity of implementing/approximating the response oracle. The authors show experimental results.

Comments:
I am afraid that the main technical result is already known:

Yoav Freund Robert E. Schapire: Adaptive game playing using multiplicative weights, Games and Economic Behavior, 29:79-103, 1999.

The paper shows that a multiplicative update algorithm can approximately solve the min-max game. If you use the result, you can readily obtain the main results of the present paper.

After rebuttal:
I read the authors comments and I understood the technical contribution more and raised my score.  Implementing/appriximating the response oracle is non-trivial. For MWU, I still think that the above paper should be cited (citing the Adaboost paper is not enough) since the paper shows MWU solves the min-max game.

---

> ### Author Response · Authors · 2018-11-25
> **This review breaks an all time record of the theater of the absurd**
>
> From the introduction: "It is well known that strategies at equilibrium in a zero sum game can be obtained by applying the celebrated Multiplicative Weights Update framework, given an oracle that computes a best response to a randomized strategy. The main technical challenge we address pertains to the characterization and implementation of such oracles."
>
> You might as well have written that the result is is already known and proven in:
>
> "John Von Neumann and Oskar Morgenstern, Theory of Games and Economic Behavior, 1953"
>
> where the authors show that every zero-sum game has a mixed equilibrium.

---

### Official Review · AnonReviewer2 · 2018-11-02
**Well-motivated approach to an interesting problem**

**Rating:** 6
**Confidence:** 4

**Review:**

This paper is concerned with the problem of finding adversarial examples for an ensemble of classifiers. This is formulated as the task of finding noise vectors that can be added to a set of examples in such a way that, for each example, the best ensemble element performs as badly as possible (i.e. it’s a maximin problem).

This is formulated as a two-player game (Equation 1), in which the above description has been relaxed slightly: Equation 1 seeks a *distribution* over noise vectors, instead of only one. This linearizes the game, so that we can seek a mixed Nash equilibrium. Given access to a best response oracle, Algorithm 1 results in such a mixed Nash equilibrium. This is pretty standard stuff (see e.g. “Robust Optimization for Non-Convex Objectives” in NIPS’17, or “A Reductions Approach to Fair Classification” in ICML’18), but the application of this approach to this problem is novel and interesting.

In Section 2.1, the authors seek to show that they can get provable guarantees for *linear* classifiers, provided that there exists a “pure strategy Nash equilibrium”, which is a set of noise vectors for which *every* classifier misclassifies *every* example. These conditions seem to me to be so strong that I’m not sure that this section is really pulling its weight.

On the subject of Section 2.1, the authors might consider whether an analysis based on “Two-Player Games for Efficient Non-Convex Constrained Optimization” (on arXiv) could be used here: convert Equation 1 into a constrained optimization problem by adding a slack variable, then reformulate it as a non-zero-sum game, in which one player uses the zero-one loss, and the other uses e.g. the hinge loss.

While Algorithm 1 makes an unrealistic oracle assumptions, and I didn’t find Section 2.1 fully satisfying, I think that overall the theoretical portion of the paper is sufficiently convincing that one should be surprised if their experiments don’t show good performance (which they do--extremely good performance, in fact). Overall, this is an interesting and important problem, and a well-motivated approach that seems to work well in practice. I think Section 2.1 is a bit weak, but this is a relatively minor issue.

---

> ### Author Response · Authors · 2018-11-25
> **The theorem gives an unconditional characterization for linear classifiers**
>
> Thanks for your review.  It seems like there is a fundamental misunderstanding about the paper and we will appreciate if you could give the paper a second read in light of our response.
>
> Your review suggests that we can only "get provable guarantees for linear classifiers provided that there exists a pure nash equilibrium."  This is not true.
>
> As we state in the introduction, page 2 in the summary of our main results:  We give a complete characterization for linear classifiers for any instance (specifically, existence of pure or mixed nash equilibrium).  In the case of a pure Nash equilibrium we prove that projected gradient descent efficiently converges to the optimal solution and otherwise finding a solution is NP-hard.  Note that the fact that finding an optimal attack even for linear classifiers is in general NP-hard is a fact, not a weakness of our proof technique.  Note also that for a modest number of classifiers as used in previous work on ensemble attacks, one can still use the characterization to find an optimal attack.
>
> Regarding the assumption that the classifiers are linear, we are not aware of principled approached for obtaining guarantees for non-linear classifiers.  The use of \alpha-approximate best response oracles in ""Robust Optimization for Non-Convex Objectives" for example avoids the problem entirely, as the whole purpose of this paper is to study the design of such \alpha approximate oracles.  Designing an approach for attacks that is provably optimal for the special case of linear classifiers is well-principled and achieves strong empirical results (as you write in your review).
>
> Specific responses to comments in the review:
>
> Regarding relationship to "Robust Optimization for Non-Convex Objectives".  This is paper is orthogonal to "Robust Optimization for Non-Convex Objectives".  In this paper we design attacks vs. defenses, but most importantly, we look at a different level of abstraction.  In "Robust Optimization for Non-Convex Objectives" the premise of the paper was to assume that one is given a blackbox oracle for solving the best-response problem.  In this paper our purpose is to design the best-response oracle;  In "Robust Optimization for Non-Convex Objectives" for linear classifiers the best response oracle was a straightforward convex optimization problem, here best response for linear classifiers is a non-convex optimization problem which requires the geometric characterization.
>
> Regarding relationship to work on “Two-Player Games for Efficient Non-Convex Constrained Optimization” (which BTW appeared on the ArXiv *after* this paper was submitted), that work seems to follow the model of "Robust Optimization for Non-Convex Objectives", and the relationship between is the same as summarized in the paragraph above.
>
> - Regarding "In Section 2.1, the authors seek to show that they can get provable guarantees for *linear* classifiers, provided that there exists a “pure strategy Nash equilibrium”, which is a set of noise vectors for which *every* classifier misclassifies *every* example. These conditions seem to me to be so strong that I’m not sure that this section is really pulling its weight." :
>
> Please see 4th bullet in introduction on page 2:
> "If the game does not have a pure Nash equilibrium, there is an algorithm for finding an
> optimal adversarial attack for linear classifiers whose runtime is exponential in the number
> of classifiers. We show that finding an optimal strategy in this case is NP-hard"
>
> Thus, this is not a limitation of our analysis, but a fact: optimal attacks, even on linear classifiers, are in general an NP-hard optimization problem.  In our paper we identify the most general sufficient condition under which optimal attacks can be carried out against a large number of classifiers (pure nash equilibirum).  With a moderate number of classifiers as has been used in literature, we are able to use the characterization to design an algorithm whose runtime is exponential in the number of classifiers and returns an optimal solution.
>
> But more importantly, as you write, this leads to a "well-motivated approach that seems to work well in practice" and "extremely good performance, in fact"
>
> Regarding: "While Algorithm 1 makes an unrealistic oracle assumptions"
>
> This is a mistake.  We do not make any assumption about the oracle.  The assumptions you may be referring to are not assumptions, but conditions on the instance and problem.  We prove our characterization using conditions on the *problem* (linear classifiers), and provide conditions on the *instance* under which efficient algorithms exist.

---

### Official Review · AnonReviewer1 · 2018-11-02
**Interesting and well-written paper but insufficient motivation**

**Rating:** 4
**Confidence:** 4

**Review:**

Summary: The authors provide a method to attack multiple classifiers, with the key insight that it is insufficient to attack a simple average of the multiple classifier outputs; creating adversarial examples which can fool each classifier independently leads to more success in attacking any defenses that has access to multiple classifiers. Note that white-box access to all the classifiers is assumed.

Clarity: Paper is well written and claims are clear and substantiated.

Originaility: The paper's technical contribution seems limited. They suggest performing PGD to estimate the best response, which is similar to previous work. However, the authors do multiple rounds of this, with different weights on the multiple classifiers at each step.

Concerns:
(1) Ensembles have mostly been proposed for black-box attacks. The setting where there are multiple classifiers and all of these weights are accessible to the attacker seems unrealistic. What's the advantage for a defense to commit to a set of trained classifiers before hand?

(2) Security concerns aside; it is not surprising that it's possible to find an attack that works for multiple classifiers at the same time, and I believe this has been done in prior work. The theoretical contribution is limited and the technique proposed is just a small modification of existing gradient based algorithms.

(3) The experimental evaluation is against previous work which tried to solve a different problem (black box based attacks).  Hence, they are not convincing.

---

> ### Author Response · Authors · 2018-11-25
> **We are not aware of previous similar work; if you wish to reject the paper due to "previous similar work"please give at least one example**
>
>
> Thank you for writing a review.  We're not able to properly respond to your review since you write that this work is similar to previous work, but without a single reference.  We are not aware of any similar work and are unable to provide proper feedback without references to alleged similar work.  We did our best to respond below.
>
>
> Specific responses:
>
> - Regarding: "Originaility: The paper's technical contribution seems limited. They suggest performing PGD to estimate the best response, which is similar to previous work."
>
> We are not aware of any previous work that proposes a principled solution to attack multiple classifiers.  In your review you write that this is "similar to previous work"; we would appreciate any reference to previous work on this so we can respond accordingly.
>
> - Regarding concern (1) "The setting where there are multiple classifiers and all of these weights are accessible to the attacker seems unrealistic."
>
> There is vast literature on whitebox attacks against a single classifier, all of which are referenced in the paper.  This paper tackles the problem of optimal attacks against multiple classifiers, for the first time.
>
> Similar to the vast literature on whitebox attacks (when all the weights are known) is interesting for two important reasons.  First, in many cases we actually do know the weights used by the classifiers.  This is simply due to the fact that many applications use existing pre-trained neural networks.  Secondly, and perhaps even more important is that even if we wish to understand how to attack classifiers that are unknown to us, we must first solve the easier problem where the classifiers are known.  That is, if we don't know how to solve the whitebox attack problem we don't know how to solve the black-box attack.
>
> The problem of how to optimally design whitebox attacks on classifiers was not known.  As we show in this paper, it is highly non-trivial, and we therefore think it makes progress on an important problem.
>
>
>
> - Regarding concern (2) "it is not surprising that it's possible to find an attack that works for multiple classifiers at the same time, and I believe this has been done in prior work."
>
> Beliefs aside, can you provide concrete references to this "prior work"?  We are not aware of any and cannot respond without a proper reference to alleged work.
>
>
> - Regarding: "The theoretical contribution is limited and the technique proposed is just a small modification of existing gradient based algorithms."
>
> Again, can you please explain which existing gradient-based algorithms you are referring to?  We are not aware of any.
>
> - Regarding (3) The experimental evaluation is against previous work which tried to solve a different problem (black box based attacks).  Hence, they are not convincing.
>
> This is not simply not true.  The experimental evaluation is against state-of-the-art methods for attacking multiple classifiers.

---

### Official Review · AnonReviewer4 · 2018-12-13
**Questions about minmax game definition**

**Rating:** 5
**Confidence:** 4

**Review:**

The paper studies the problem of adversarial examples generation. The authors phrase the following problem: given a set of models  C, we want to find an adversarial perturbation that maximizes the loss on an ensemble of models. However, the ensemble weights are chosen by the learner. In the case that we have one example, this is equivalent to asking that the same adversarial perturbation (or distribution over perturbations) fools all the individual models in my collection. This is a reasonable phrasing of the problem, though it seems different from versions studied in literature. In particular, previous works used uniform ensembles.
More generally, the authors consider a set of m examples, and the adversarial player now looks for a (distribution over) perturbations for each of the n examples. The learner player selects mixing weights to minimize the error rate. This is an interesting formulation of the problem: in particular, tying the mixing weights used for all examples is a non-intuitive change and does not have the clean interpretation above any more.
This notion of allowing mixing weights on the learner is a change from previous work. The authors would do well to explain why this formulation is chosen and what the interpretation is. It corresponds to a specific attack model where the learner and the adversary make choices in a very specific order, and could use further explanation on when this a reasonable attack model. Note that previous work looked at the setting of all weights being equal, and one natural variant is to allow a set of mixing weights per example, which would correspond to finding a perturbation (or distribution over perturbations) for this example that fool all models in the set C. The version studied here is left unexplained in the current work.

The authors then argue that we can solve this game by playing MW vs. best response. They propose using best response on the adversarial player. This player is then trying to find the perturbation that maximizes the p-weighted sum of the 0-1 (or rather surrogate) losses, where p represents the mixing weights on C. The authors show that in the convex case, if there is a pure NE, then the best response can be found: in this case we get a convex problem. They study the convex case a bit more, showing that there is at most an exponential number of values for the 0-1 loss, since a {0,1} vector defining which side of each classifier in C x falls in fully defines the loss at x.

Finally, the authors move to the non-convex case where the experiments are done. The authors report interesting results on imagenet and for mnist for the convex case. I had some trouble understanding the imagenet results. For one, it seems fishy that their Madry et al. attack is worse than the FGM for many of the models and suggests strongly that the parameters for the Madry attack were not properly tuned. It is hard to know since the paper does not report on various parameters for these attacks. Second, these attacks are designed for l_infty and modifying them for l_2 would be necessary for a fair comparison. Finally, I am not sure why the authors do not compare to the Carlini and Wagner attacks on Imagenet, which is actually an l2 attack and makes the accuracy 0 at a slightly larger perturbation radius. Also, the authors would do well to emphasize that for larger perturbation radii, there are attacks which make the accuracy zero, and the contribution here seems to be look at smaller radii.

My primary concern with the work is that it is not clear to me how the specific two player game is motivated. The authors do not justify why it makes sense to allow weights on the ensemble, and also why these weights need to be tied together across examples. For a paper that makes strong claims about its approach being principled, this is a serious shortcoming in my view. Secondarily, the experiments section leaves me worried that the comparison is with improperly tuned versions of previous work. I would therefore not be in favor of accepting this paper.

Comments:
pg 1 : "One of the most pressing ..." : that is perhaps an unnecessary exaggeration.
pg 2: The name "NSFW" is an unfortunate choice, is completely non-informative about the contribution and I strongly recommend the authors reconsider it.
- As far as I can tell, Tramer et al. do not build an ensemble model at all; the ensemble word there refers to an ensemble of adversarial perturbations.
- The hinge loss is actually not smooth. However, I don't quite see why you need smoothness there.

---

> ### Author Response · Authors · 2018-12-13
> **Thanks for the review**
>
> We address all the concerns below.
>
> The review expresses questions about the zero sum game formulation (why there is use of weights and why the problem is formulated for multiple points), and does not find the parameters of the experiments.  This can all be found in the paper and we refer the reviewer to places in the paper where we explain why weights are used, attacks on multiple points, and refer the reviewer to the sections where we precisely describe the parameters of the experiments.
>
> We'd appreciate if you read our response and consider reevaluating your score accordingly.
>
> 1. "it is not clear to me how the specific two player game is motivated. The authors do not justify why it makes sense to allow weights on the ensemble, and also why these weights need to be tied together across examples."
>
> - with regards to: " The authors do not justify why it makes sense to allow weights on the ensemble"
>
> The weights are the probabilities that the learner assigns to the classifier, as stated in the first page of the introduction where we write:
>
> "Furthermore, a learner can randomize over classifiers and avoid deterministic attacks (see Figure 1)".  In the figure we illustrate and describe an example where randomization gives a learner power to avoid adversarial attacks.
>
> - with regards to "why these weights need to be tied together across examples"
>
> The weights do not need to be tied together across examples.
>
> In the paper we defined the attack for m data points, *for any value of m* and one can apply all the results for m=1.  Note that all our results are independent of the number of data points m and are a function of the number of classes k, number of classifiers n, noise budget \alpha, and approximation \delta.  The parameter m does not affect the complexity of the problem.
>
> Given m data points, one can construct an adversarial attack for every single data point using the framework described here.  The formulation of the zero sum games remains identical: the learner randomizes over n classifiers and the learner randomizes over noise vectors.
>
>
> 2. With regards to your comments on the experiments,
>
> “For one, it seems fishy that their Madry et al. attack is worse than the FGM for many of the models and suggests strongly that the parameters for the Madry attack were not properly tuned. It is hard to know since the paper does not report on various parameters for these attacks. “
>
> We provide full details on the experimental setup in the appendix. In the first paragraph, we clearly state how we use the same choice of parameters that the authors employed in their original paper: “When running the Projected Gradient Method by Madry et al., given a noise budget α, we run the algorithm for 40 iterations with a step size of α/40 × 1.25 so as to mimic the setup of the authors”
>
> “Second, these attacks are designed for l_infty and modifying them for l_2 would be necessary for a fair comparison”
>
> As stated in the paper, we use the cleverhans implementation of these attacks. The library, maintained by Goodfellow and Papernot, provides an option to use the Madry attack as designed for the L2 norm which only involves a projection to the L2 ball instead of L infinity.
>
> “Finally, I am not sure why the authors do not compare to the Carlini and Wagner attacks on Imagenet, which is actually an l2 attack”
>
> The Carlini Wagner attack is based off a Lagrangian formulation and is hence not guaranteed to return solutions that lie within a prespecified noise budget. While we could clip solution to lie within a specific range, this is not the way the algorithm was designed to be used and would be an unfair comparison. Hence, we compare against other methods, such as the Momentum Iterative Method (the strongest attack as per the 2017 NIPS competition) that explicitly allow for noise constraints.
>
>
>
> 3. Regarding “The hinge loss is actually not smooth. However, I don't quite see why you need smoothness there.“
>
> The hinge loss is smooth, this can be verified from the definition. The condition is necessary to ensure the faster convergence rate of our theorem.

---

> > ### Comment · AnonReviewer4 · 2018-12-13
> > **Thanks for the response**
> >
> > Thanks for the quick response.
> >
> > "The weights are the probabilities that the learner assigns to the classifier, as stated in the first page of the introduction where we write"
> >
> > Let me try to be clearer. Previous work asked : "suppose I want an adversarial perturbation that fools each of k classifiers. How do I find one". You are asking a different question from this, perhaps without realizing it. Your question translates to "Suppose I want a distribution over adversarial perturbations that fools each of k classifiers with some probability".  When you change the question, you have to justify why this is an interesting question to ask. The old question, e.g.,  helps answer "Suppose as a learner I were to apply all of the models on an image and only label it if they all agree. How resistant is this ensemble?". Since in your model, each classifier is seeing a different sample from the adversarial perturbation,  one question being answered is "suppose as a learner, I were to pick one classifier and apply it on the image to label it, how resistant can I make that if I picked this classifier from a distribution (or in response the adversary's randomized strategy)?". This is a very different question. Your example shows why the answer can be different. The paper does not make a case that the question is interesting.
> >
> > "The weights do not need to be tied together across examples.  "
> > Your paper would be improved if your equations said so as well.
> >
> > "we run the algorithm for 40 iterations with a step size of α/40 × 1.25 so as to mimic the setup of the authors”
> > Likely minor point: If I am calculating correctly, the 1.25 should be a 1.33 (they had alpha = 0.3, step-size = 0.01).
> >
> > I find it a very counterintuitive that PGD is worse than FGM. This might suggest that something strange happens when you play against ensembles, or may suggest that the parameters weren't quite optimized correctly in your implementation (e.g. the optimal "1.25/1.33" may be different for l2 as compared to linfty).
> >
> > "The hinge loss is smooth, this can be verified from the definition."
> > The hinge loss is l_h(w; x, y) = [f(w; x,y)]_+. When f is -eps, the gradient is zero. When f is plus eps, the gradient is non-zero and bounded below. This violates smoothness, which would require that the gradient be Lipschitz.

---

### Meta-Review · Area_Chair1 · 2018-12-17
**Unclear motivation and significance of empirical results**

**Confidence:** 3
**Recommendation:** Reject

**Metareview:**

Four reviewers have evaluated this paper. The reviewers have raised concerns about the specific formulation used for adversarial example generation which requires further clarity in motivation and interpretation. The reviewers have also made the point that the experimental evaluation is against previous work which tried to solve a different problem (black box based attack) and hence the conclusions are unconvincing.